# Investigating the accuracy of blood oxygen saturation measurements in common consumer smartwatches

**Yihang Jiang**[1], **Connor Spies**[2], **Justin Magin**[2,3], **Satasuk Joy Bhosai**[2,4], **Laurie Snyder**[2,4], **Jessilyn Dunn**[1] *

**1** Department of Biomedical Engineering, Duke University, Durham, North Carolina, United States of America, **2** Department of Medicine, Duke University, Durham, North Carolina, United States of America, **3** School of Medicine, The University of North Carolina at Chapel Hill, North Carolina, United States of America, **4** Duke Clinical Research Institute, Duke University, Durham, North Carolina, United States of America

* jessilyn.dunn@duke.edu

## Abstract

Blood oxygen saturation (SpO2) is an important measurement for monitoring patients with acute and chronic conditions that are associated with low blood oxygen levels. While smartwatches may provide a new method for continuous and unobtrusive SpO2 monitoring, it is necessary to understand their accuracy and limitations to ensure that they are used in a fit-for-purpose manner. To determine whether the accuracy of and ability to take SpO2 measurements from consumer smartwatches is different by device type and/or by skin tone, our study recruited patients aged 18–85 years old, with and without chronic pulmonary disease, who were able to provide informed consent. The mean absolute error (MAE), mean directional error (MDE) and root mean squared error (RMSE) were used to evaluate the accuracy of the smartwatches as compared to a clinical grade pulse oximeter. The percent of data unobtainable due to inability of the smartwatch to record SpO2 (missingness) was used to evaluate the measurability of SpO2 from the smartwatches. Skin tones were quantified based on the Fitzpatrick (FP) scale and Individual Typology Angle (ITA), a continuous measure of skin tone. A total of 49 individuals (18 female) were enrolled and completed the study. Using a clinical-grade pulse oximeter as the reference standard, there were statistically significant differences in accuracy between devices, with Apple Watch Series 7 having measurements closest to the reference standard (MAE = 2.2%, MDE = -0.4%, RMSE = 2.9%) and the Garmin Venu 2s having measurements farthest from the reference standard (MAE = 5.8%, MDE = 5.5%, RMSE = 6.7%). There were also significant differences in measurability across devices, with the highest data presence from the Apple Watch Series 7 (88.9% of attempted measurements were successful) and the highest data missingness from the Withings ScanWatch (only 69.5% of attempted measurements were successful). The MAE, RMSE and missingness did not vary significantly across FP skin tone groups, however, there may be a relationship between FP skin tone and MDE (intercept = 0.04, beta coefficient = 0.47, p = 0.04). No statistically significant difference was found between skin tone as measured by ITA and MAE, MDE, RMSE or missingness.

**Data Availability Statement:** Deidentified data used for submission is available at: https://github.com/DigitalBiomarkerDiscoveryPipeline/Digital_

Health_Data_Repository/tree/main/Blood%
20Oxygen%20Saturation%20(SpO2).

**Funding:** This study was supported in part by funds from AstraZeneca (LS and SJB) and by the National Institute of Diabetes and Digestive and Kidney Diseases of the National Institutes of Health (R01DK133531 to JD). The funders had no role in study design, data collection and analysis, decision to publish, or preparation of the manuscript.

**Competing interests:** I have read the journal's policy and the authors of this manuscript have the following competing interests: JD is a scientific advisor to Veri, Inc. SJB is a founder of Pluto Health. All other authors declare no competing interests.

## Author summary

Blood oxygen saturation (SpO2) is an important measurement for monitoring patients with acute and chronic conditions that are associated with low blood oxygen levels, including chronic obstructive pulmonary disease (COPD), heart failure, asthma, and pneumonia, among other conditions. While smartwatches may provide a new method for continuous and unobtrusive SpO2 monitoring, it is necessary to understand their accuracy and limitations to ensure that they are used in a fit-for-purpose manner.

## Introduction

Pulse oximetry technology is an easy, painless measure of peripheral oxygenation. Pulse oximetry has been in existence since 1974 and has revolutionized the ability to monitor acute changes as well as chronic diseases that may affect blood oxygen saturation levels ($SpO_2$) to determine both needed interventions and to assess if those interventions are effective. It is widely used both for ease and accessibility but also because chronic lung disease is one of the most common health problems worldwide and the third leading cause of death, with almost 545 million people reporting a chronic respiratory condition in 2017, a ~40% increase from 1990 [1]. Seasonal, weekly, and daily variability of Chronic obstructive pulmonary disease (COPD) symptoms are found in 60%, 54% and 44% of patients respectively [2], which makes pulse oximetry necessary for both home monitoring and in hospital settings to monitor how oxygen saturation changes acutely with symptoms like breathlessness.

Pulse oximetry technology leverages the difference in light absorption between deoxygenated and oxygenated hemoglobin to monitor oxygen saturation. This technology is considered to be a reliable alternative to the more invasive arterial blood gas measurement, which requires trained professionals and specialized analysis equipment. Pulse oximetry has come front and center during the COVID-19 pandemic in two ways. First, COVID-19 infection can lead to oxygen desaturations, which either may not be evident early in the infection for individuals at home [3] or may require new oxygen supplementation at hospital discharge [4,5], such that home monitoring of oxygen saturation has become a critical component of COVID-19 recovery at home. Second, with the shift to more telemedicine visits during the pandemic, individuals with chronic respiratory conditions were increasingly relying on at-home monitoring, which included pulse oximetry [6]. As a result, lung disease, either as a consequence of COVID-19 infection or due to the limited in-person evaluations because of the COVID-19 pandemic, are now increasingly managed in home settings [7].

In parallel to the changing clinical needs for home monitoring, pulse oximetry has been integrated into multiple common consumer smartwatches. In fact, the pulse oximetry feature of smartwatches has become a major selling point which has likely contributed to the global market value of smartwatches to reach $33.1 billion in 2022 [8]. However, evidence of the performance of these devices remains limited, and disclaimers may be overlooked by consumers (S1 Text) [9]. The few studies that do exist do not investigate the effects of individual characteristics (e.g., skin tone), compare across multiple devices or thoroughly examine performance in low ranges, or explore patterns of error and bias (e.g., consistent over- or underestimation of measurements) or missingness [10–13]. Along these lines, we recently tested the accuracy of heart rate measurements from commercial smartwatches and found that, while there was no difference in accuracy based on skin tone, during physical activity the error increased by 30% [14]. Further, the accuracy and data missingness varied substantially by device type.

There is existing discussion surrounding pulse oximetry inaccuracies and increased smartwatch inaccuracies and data missingness under certain conditions [14–19]. For example, racial bias can lead to undetected hidden hypoxemia. Results from a large study [17] that includes 1333 white patients and 276 black patients indicated that the incidence of hypoxemia can be 3 times higher among patients self-reported as Black vs White because the oxygen saturation readings of black patients tend to be overestimated at a low SpO2 range. We hypothesized that smartwatches would demonstrate a range of accuracy that may be influenced by skin tone. To test this hypothesis we compared four commercially available smartwatches against a clinical-grade pulse oximeter in a range of patients with and without respiratory conditions. We chose these devices based on their advertised ability to capture real-time measurements of oxygen saturation. We also utilized a handheld colorimeter to measure skin tones in a standardized manner. To our knowledge, this is the first study to explore the accuracy of blood oxygen saturation measurements from consumer smartwatches as well as patterns of missingness, error, and bias in patients likely to experience intermittent oxygen desaturations.

## Results

### Patient cohort

A total of 49 individuals were enrolled in and completed the study, which included 31 males and 18 females. The racial breakdown was 34.7% Black and 65.3% White. The median age was 64 (range, 38–76) and the median Body Mass Index (BMI) was 28.8 kg/m$^2$ (range, 21.4–42.8 kg/m$^2$). Of these participants, 30 were recruited from outpatient clinics, and 19 were inpatient. Two participants did not have any disease-causing hypoxemia, while 42 had pulmonary disease, three had cardiovascular disease, and two had both pulmonary and cardiovascular disease. For those with pulmonary disease, 28 (64%) had interstitial lung disease, 12 (27%) had chronic obstructive pulmonary disease, and four (9%) had pulmonary hypertension. For those with cardiovascular disease, the primary diagnosis for all five (100%) was heart failure. Additionally, 21 subjects were lung transplant recipients, one was a heart transplant recipient, and one was a heart and lung transplant recipient. Of the 49 participants, 16 were receiving oxygen via a nasal cannula, with a median of three liters per minute, and one was receiving oxygen via a non-rebreather mask at 15 liters per minute. Participants ranged from Type II to Type VI on the Fitzpatrick skin tone scale, with an average of 10 participants in each category. On the ITA skin tone scale, participants ranged from -56˚ to 53˚, with a median value of 21˚. There was a strong correspondence between the skin tone values assessed by these two measurement modalities, with a Spearman correlation coefficient between the FP skin tone group and the objective ITA measurement value of 0.928 (S1 Fig). Of the 49 total participants, 11 were missing skin tone measurements on the ITA scale because we did not yet have the device used to measure skin tone objectively (S4 Table). Overall, 588 blood oxygen saturation measurements were attempted, 451 were made successfully, with an average of 2.3 measurements/device/patient.

### Device performance

#### Smartwatch pulse oximetry measurement accuracy and usability of generating measurements

The Masimo MightySat Rx SpO$_2$ measurements ranged from 82.0% to 100.0%, with an average of 95.8% and a median of 97.0% (S1 Table). Average SpO$_2$ values from the Apple Watch Series 7, Garmin Venu 2s, Garmin Fenix 6 Pro, and Withings ScanWatch devices were 96.3%, 90.4%,

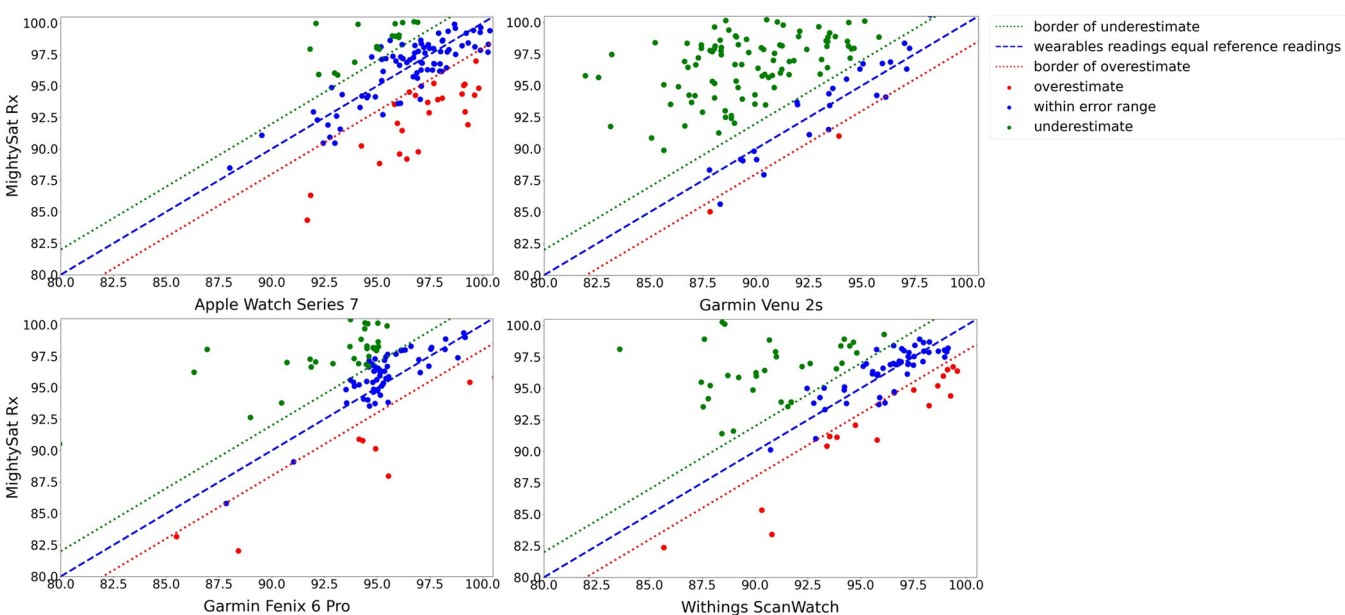

**Fig 1.** Accuracy of SpO$_2$ measurements against the FDA-cleared Masimo MightySat reference standard for A) Apple Watch Series 7, B) Garmin Venu 2S, C) Garmin Fenix 6 Pro, and D) Withings ScanWatch. Data falls into the categories of overestimate (red), within error range, meaning the difference of readings between reference reading and device reading is within a 2% error range, (blue), or underestimate (green). The dotted lines on the graph represent the 2% error of the clinical-grade MightySat Rx. Additionally, the relative percentage of data falling into the categories of overestimate (red), within error range (blue), underestimate (green), and missing (gray) are summarized in Fig 1.

94.4%, and 94.3%, respectively. Each smartwatch measurement was compared to the concomitant Masimo MightySat Rx reference measurement to determine whether it fell within, above, or below the reference device accuracy range (Fig 1; Table 1). The Apple Watch Series 7 had the highest percentage of readings (58.3%) falling within the accuracy range of the reference device, and also had the highest percentage of overestimated measurements (17.4%), meaning that the watch reported a higher blood oxygen saturation than the clinical-grade oximeter value plus a 2% error. The Garmin Venu 2s had the highest percent of underestimated readings (67.4%), wherein the device-reported SpO$_2$ value was lower than the clinical-grade pulse oximeter SpO$_2$ value minus a 2% error (Fig 2).

The Withings ScanWatch and the Garmin Fenix 6 Pro had the highest data missingness of the devices tested, indicating poor reliability for successfully obtaining a measurement when a measurement was attempted. Of the total number of expected measurements, 31% and 28% of the values for the Withings ScanWatch and the Garmin Fenix 6 Pro, respectively, were blank measurements. The missingness of the Apple Watch, Garmin Venu 2s, Garmin Fenix 6 Pro, and Withings device data ranged from 11%, 14%, 28%, and 31%, respectively, which indicates

**Table 1. Percentage and number of measurements falling in four categories.**

|  | No. (%) of overestimated SpO$_2$ values | No. (%) of within error range | No. (%) of underestimated SpO$_2$ values | No. (%) of missed measurements |
|---|---|---|---|---|
| Apple Watch Series 7 | 25 (17) | 84 (58) | 19 (13) | 16 (11) |
| Garmin Venu 2s | 2 (1) | 25 (17) | 97 (67) | 20 (14) |
| Garmin Fenix 6 Pro | 8 (6) | 58 (42) | 33 (24) | 39 (28) |
| Withings ScanWatch | 16 (11) | 51 (35) | 33 (23) | 44 (31) |

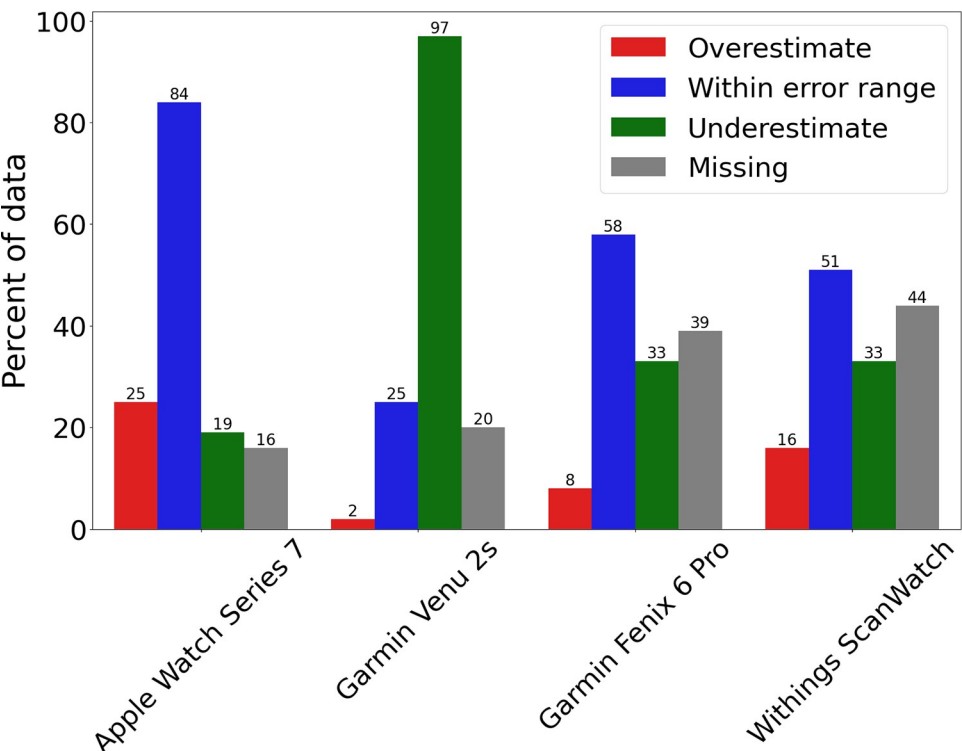

**Fig 2. Relative percentages of data falling into the categories of overestimate (red), Within error range (blue), underestimate (green), and missing (grey) for the Apple Watch Series 7, Garmin Venu 2S, Garmin Fenix 6 Pro, and Withings ScanWatch.** The black numbers above each bar represent the number of measurements within the category.

the proportion of time in this study when the device was unable to generate measurements (Table 1).

## Comparison of accuracy and reliability of generating measurements across device types

To further quantify potential differences between the wearables by skin tone, we analyzed the MAE, MDE, and RMSE by device type and by FP skin tone group. The MAE of each device ranged from 2.2% to 5.8% SpO$_2$, with lower MAE indicating higher accuracy. The Apple Watch Series 7 had the lowest MAE and the Garmin Venu 2 had the highest MAE. The MDE ranged from -0.4% to 5.5% SpO$_2$, with MDE values closer to zero indicating higher accuracy. The Apple Watch Series 7 had the MDE closest to zero and the Garmin Venu 2 had the MDE farthest from zero. The RMSE ranged from 2.9% to 6.7% SpO$_2$, with lower RMSE indicating higher accuracy. The Apple Watch Series 7 had the lowest RMSE and Garmin Venu 2 had the highest RMSE. (Fig 3).

To explore whether statistically significant differences exist among MAE, MDE, RMSE, and missingness across the different commercial wearables, we used two-sided, paired t-tests with Bonferroni multiple hypothesis correction. The Bonferroni threshold for comparison of MAE, MDE, RMSE, and missingness was 0.0083. Using this threshold, the Garmin Venu 2s demonstrated a statistically significantly higher MAE and RMSE as compared with all other devices. For MDE, every pairwise comparison showed a significant difference except for the Garmin Fenix 6 compared to the Withings ScanWatch, indicating their similarity in performance along this metric. This indicates that directionality of errors may serve as a useful metric for

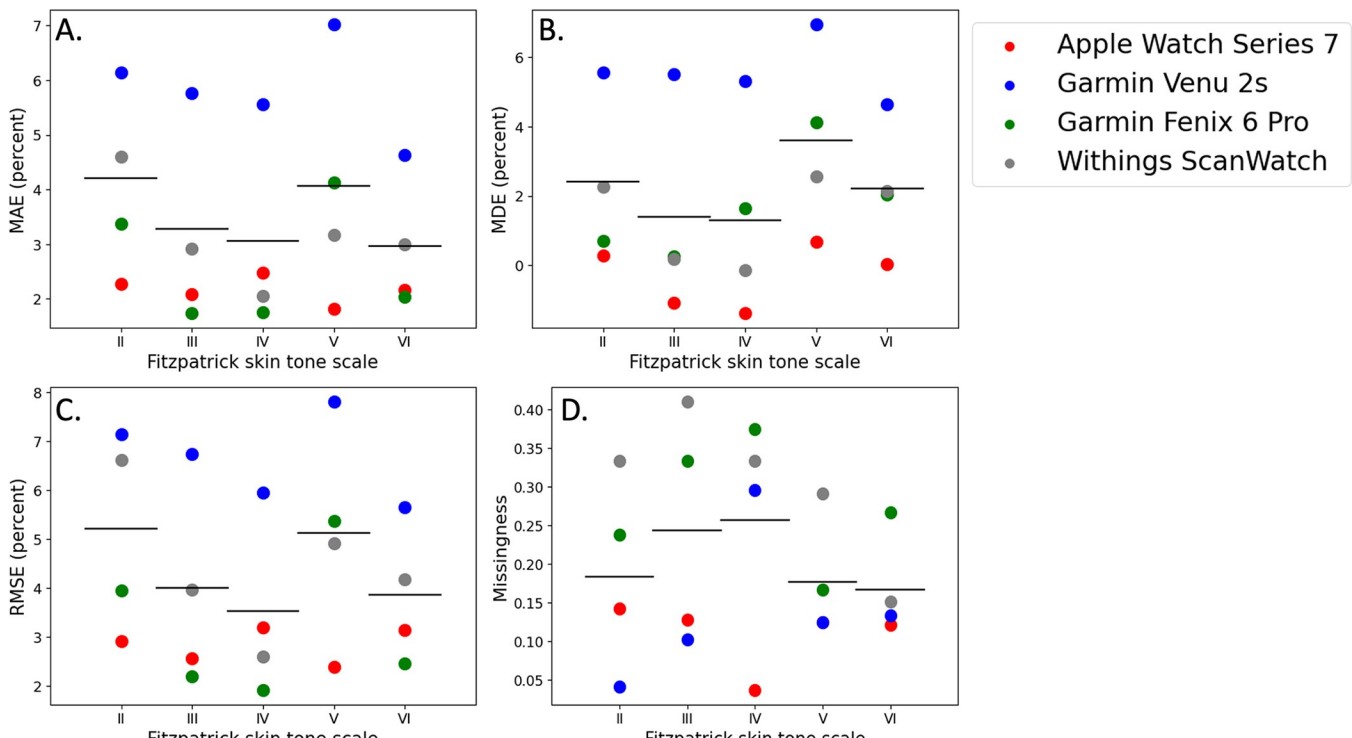

**Fig 3.** (A) Mean Directional Error (MDE) (B) Mean Absolute Error (MAE) (C) Root Mean Square Error (RMSE) (D) Missingness across skin tones classified by Fitzpatrick scale.

comparing across different smartwatch-based oxygen saturation measurement technologies. The Apple Watch Series 7 had significantly lower missingness compared with both the Garmin Fenix 6 and the Withings ScanWatch, both of which had similarly high missingness (Table 1).

*Comparison of accuracy and reliability of generating measurements across skin tones.* The FP skin tone of participants ranged from group II to group VI (Fig 4). The MAE for each skin tone group was 4.2%, 3.3%, 3.1%, 4.1%, 3.0% from FP II to FP VI, respectively. Notably, FP IV had the lowest MAE and FP II had the highest MAE. The MDE for each skin tone group was 2.4%, 1.4%, 1.3%, 3.6%, 2.2% from FP II to FP VI, respectively. The MDE closest to zero was for group FP VI and the MDE farthest from zero was for FP V. The missingness for each skin tone group was 18.4%, 24.4%, 25.7%, 17.7%, 16.7% from FP II to FP VI, respectively. The RMSE for each skin tone group was 5.2%, 4.0%, 3.5%, 5.1%, 3.9% from FP II to FP VI, respectively. The RMSE closest to zero was for group FP IV and the MDE farthest from zero was for FP II.

Neither MAE, MDE, RMSE, nor missingness demonstrated an apparent relationship to skin tone. To explore whether statistically significant differences exist among MAE, MDE, RMSE, and missingness across different skin tone groups, we used two-sided, unpaired t-tests with Bonferroni multiple hypothesis correction (Table 2). The Bonferroni threshold for comparison of MAE, MDE, RMSE and missingness was 0.005. For all pairwise comparisons of these four metrics across the different FP skin tone groups, no statistically significant difference was found.

### Relationships between device performance and skin tone

To explore whether there is a statistically significant relationship between skin tone (as measured by either the FP or Delfin ITA scale) and MAE, MDE, RMSE, or missingness, we fit

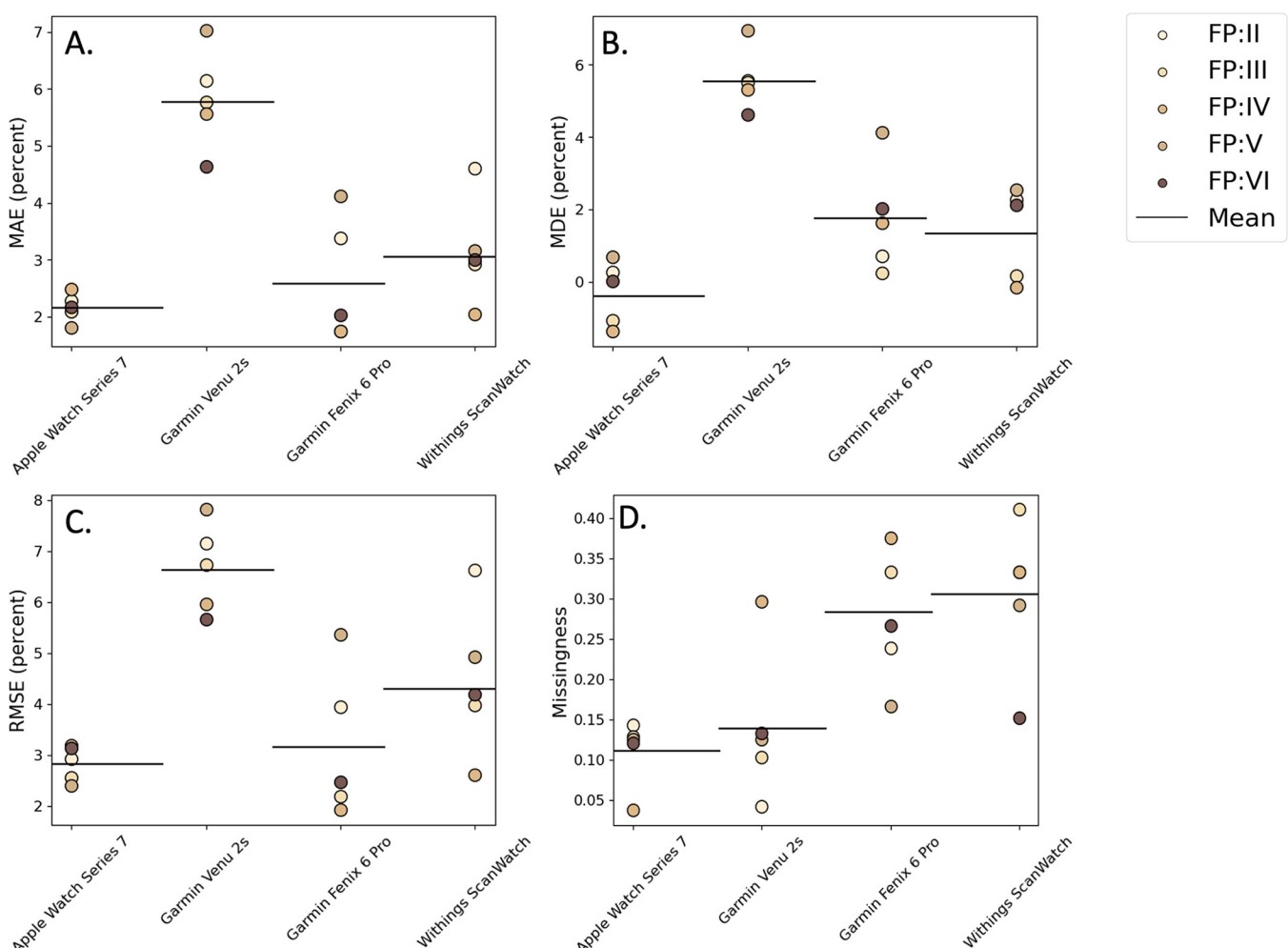

**Fig 4.** (A) Mean Absolute Error (MAE), (B) Mean Directional Error (MDE), (C) Root Mean Square Error (RMSE) and (D) Missingness across four commercial wearable devices by FP skin tone group.

an ordinary least squares model (p-values of 0.88, 0.04 and 0.36, respectively, with coefficient values listed in S2 Table). Of MAE, MDE, RMSE, and missingness, we found that MDE may have a relationship with the FP skin tone groups (intercept = 0.04, beta coefficient = 0.47, p = 0.04). However, in using the ITA measurement scale, no statistically significant difference was seen (S3 Table). It should be noted that 11 of 49, or 22% of the study population, were missing ITA values, reducing the robustness of these findings as compared with the FP findings.

**Table 2. P-values from two-sided paired t-tests with Bonferroni multiple hypothesis correction.**

|  | Apple vs Venu | Apple vs Fenix | Apple vs Withings | Venu vs Fenix | Venu vs Withings | Fenix vs Withings |
|---|---|---|---|---|---|---|
| MAE | 9.52e-09 | 0.10 | 0.010 | 9.08e-08 | 0.00015 | 0.52 |
| MDE | 3.19e-17 | 0.00011 | 0.00088 | 3.09e-11 | 4.56e-08 | 0.56 |
| RMSE | 1.34e-07 | 0.10 | 0.019 | 3.94e-07 | 0.00053 | 0.21 |
| Missingness | 0.60 | 0.0050 | 0.0022 | 0.022 | 0.022 | 0.71 |

## Materials and methods

### Study population

This study was conducted under the approval of the Duke University Health System (DUHS) Institutional Review Board and all participants signed informed consent before participating (Pro00105579). Criteria for study inclusion were patients 18–85 years old, with and without respiratory disease, who were receiving care at Duke Hospital and Clinics, a tertiary referral center. Additionally, we specifically recruited individuals with a wide range of skin tones. Exclusion criteria included patients with peripheral vascular disease, Raynaud's syndrome, cryoglobulinemia or any collagen vascular disease affecting the fingers, a history of blood clots in the last 6 months, an essential tremor, or gel nail polish or any other non-natural, non-removable discoloration of the forefinger.

### Devices and data collection

We utilized four consumer smartwatches that offered pulse oximetry and heart rate (HR) sensors that could be manually triggered to produce measurements. These included the Apple Watch Series 7, Garmin Venu 2s, Garmin Fenix 6 Pro, and Withings ScanWatch (Fig 5A). These watches were compared against the Masimo MightySat Rx, a clinical-grade finger pulse oximeter reference standard currently not thought to be affected by racial bias[20], with a reported 2% error measurements without motion (Fig 5B) and also compared the

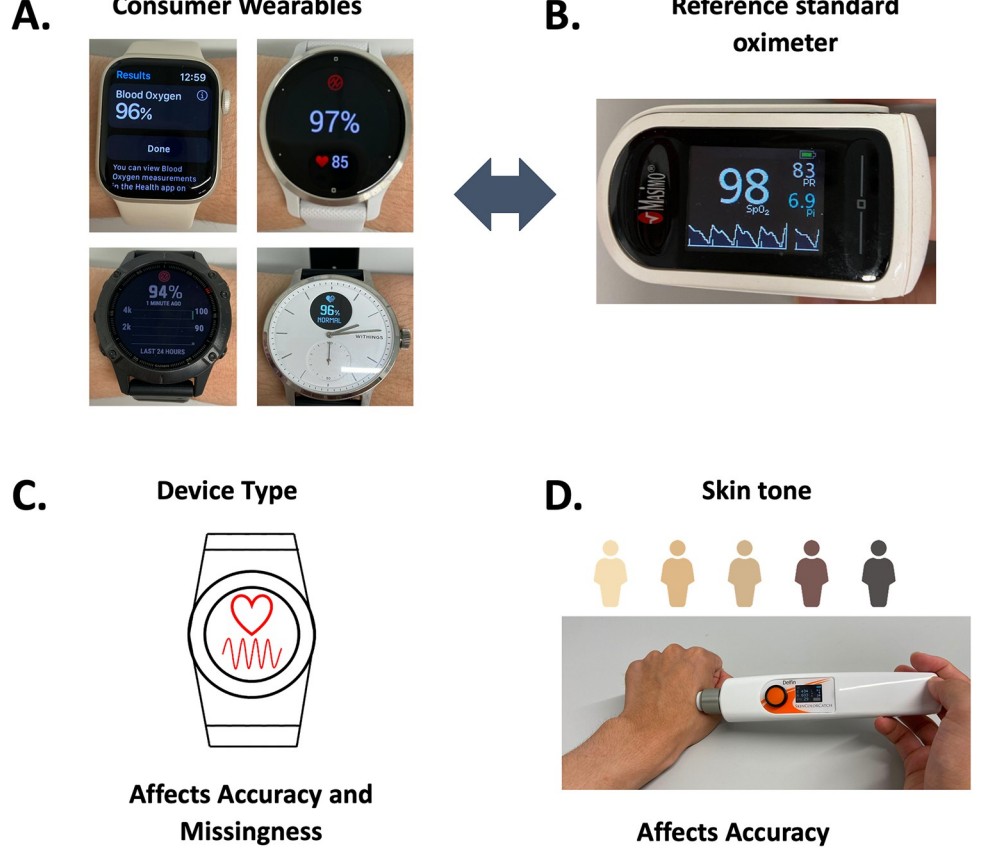

**Fig 5.** Graphical abstract of the study, including the four devices that were tested (A), the reference standard (B), the role of device type on the accuracy and missingness of SpO2 values (C), and the role of skin tone on the accuracy of SpO2 values (D).

smartwatches against one another [21]. We measured skin tone initially by the Fitzpatrick scale (FP) and later added the Delfin colorimeter that uses RGB color sensors to receive light reflected from the surface of skin. Upon enrollment, patients' skin tone was measured on the ITA scale using the Delfin SkinColorCatch, and visually on the FP scale (Fig 5D). Due to delays in obtaining the Delfin SkinColorCatch device, the 11 initial patients only have skin tone measurements on the FP scale. Additionally, patients' body mass index, age, race, sex, and medical history were obtained from their most recent electronic health record entry.

For the study procedure, the smartwatches were placed tightly on the wrist, approximately one centimeter above the ulnar styloid process. The Masimo finger pulse ox was placed on the middle finger of the subject's dominant hand. The measurements were taken in two rounds, first with the Apple Watch Series 7 and Garmin Venu 2s on separate and randomized wrists, followed by the Garmin Fenix 6 Pro and Withings ScanWatch on separate and randomized wrists. In each round, three measurements of HR and $SpO_2$ were captured from each watch and the pulse oximeter while the participant was at rest, with approximately three minutes between measurements. The patients were asked to keep still and were in the seated or reclining position during measurements to eliminate the difference between the dominant hand and non-dominant hand. During the study procedure, if a smartwatch failed to produce a measurement, it was readjusted and repositioned on the subject's wrist. If the device still failed to produce a measurement, no value was recorded for this specific time point and it was labeled as a missing (unobtainable) observation.

## Metrics and statistical analysis

The metrics of mean absolute error (MAE) (Eq 1), mean directional error (MDE) (Eq 2), percent of data not obtainable (i.e., missingness, Eq 3) and root mean square error (RMSE) (Eq 4) were used to evaluate the accuracy and usability of the $SpO_2$ measurement function from the smartwatch devices. Two-sided unpaired t-tests were used to compare the MAE, MDE, missingness and RMSE among different skin tones and two-sided paired t-tests were used to compare these same metrics among different wearable devices with Bonferroni multiple hypothesis correction. Ordinary Least Squares Linear Regression was applied to determine whether there was any relationship between the assessment metrics MAE, MDE, missingness, and RMSE, and skin tones (measured based on the FP and ITA scales).

$$\text{MAE (Mean absolute error)} = \frac{\sum |\text{Detected SpO}_2 - \text{Reference SpO}_2|}{\text{Number of valid measurements}} \tag{1}$$

$$\text{MDE (Mean directional error)} = \frac{\sum (\text{Reference SpO}_2 - \text{Detected SpO}_2)}{\text{Number of valid measurements}} \tag{2}$$

$$\text{Missingness}(\%) = 100 - \left(\frac{\text{Valid measurements}}{\text{Total measurements}}\right) * 100 \tag{3}$$

$$\text{RMSE (Root Mean Square Error)} = \sqrt{\frac{\sum (\text{Detected SpO}_2 - \text{reference SpO}_2)^2}{\text{Number of valid measurements}}} \tag{4}$$

## Discussion

The oxygen saturation measurement functionality has been adopted by multiple smartwatch manufacturers as a desirable feature and selling point. Here, we explored four representative

commercial wearables with high market share [22,23]. We compared their readings with the readings from a reference standard [24] pulse oximeter, the Masimo MightySat Rx, across study participants with a range of different skin tones in order to compare the blood oxygen saturation measurement feature of multiple commercial wearables and explore skin tone as a potential moderator of their performance.

When comparing the oxygen saturation detection function of different devices, it is critical to report multiple performance metrics to give a complete representation of the performance, as a single metric can fail to reveal differences in performance even when they do in fact exist. In this study, if only MAE was reported, we would conclude that the Apple Watch, Garmin Fenix Pro 6, and Withings ScanWatch all have similar performance. However, when MDE is reported, we find that there is a significant difference between these devices. The reason is that MAE only measures the average magnitude of errors but ignores the directionality of the errors, meaning the proportion of underestimated and overestimated measurements. Overestimated measurements are potentially dangerous if a person decides not to seek care as a result of considering the inaccurate measurement. Many digital health devices make claims that meet FDA standards, but this does not always mean that they have been rigorously validated. Simply, it could mean that a device is not harmful, but not necessarily accurate or backed by sufficient clinical data [25]. Conversely, some devices may err on the side of underestimation in order to avoid the potentially dangerous situations that can occur when a measurement is overestimated, but this situation can lead to alarm fatigue if measurements are consistently underestimated, leading users to ignore the measurements altogether.

COVID-19 has accelerated the wide interest in remote patient monitoring. Pulse oximetry can be an effective tool to detect the risk of deterioration and promote patient safety. The advent of smartwatch-based oxygen saturation monitoring tools opens up new opportunities for remote and continuous patient monitoring, and it is necessary that this technology continues to be improved upon to increase its accuracy, usability, and equitability.

Skin tone can be quantified in multiple ways, including both subjective and objective, as well as continuous and discrete methods. In our study, we used two separate methods to assess skin tone: the objectively-measured continuous ITA value reported by the Delfin SkinColor-Catch, and the subjective ordinal Fitzpatrick scale value as recorded by our clinical research specialist. While the ITA and Fitzpatrick measurements demonstrated a strong correlation, the continuous ITA values are difficult to employ as compared with the ordinal Fitzpatrick measurements for the statistical analyses presented herein. Larger studies with more power would benefit from employing the objective and continuous numerical value corresponding to skin tone to eliminate subjectivity in both the chosen scale and in the act of measuring. The arterial blood gas (ABG) test is a blood test that requires samples from a patient's artery to measure the levels of oxygen and carbon dioxide in the blood. Since we didn't use arterial blood gas to calculate oxygen saturation, it is possible that racial bias exists in our reference gold standard, which can affect the accuracy of subjects with dark skin tones. In future studies, this more invasive but "gold standard" ABG test can be employed to avoid potential confounding effects of skin tone on $SpO_2$ measurements that may affect pulse oximetry.

The Fitzpatrick scale was developed based on self-reported erythema sensitivity and the ability of Caucasians to tan, and has been commonly used to categorize skin tones into six categories [26]. It's original purpose was not to measure skin tone, but rather to assess skin's reaction to sunlight. Further, the inter-rater reliability of dermatologists is known to be suboptimal, particularly for darker tones between III and IV and IV and V [27]. As such, the Fitzpatrick scale is not an ideal tool for objective skin tone measurements, and thus in our study, we also employ a more objective optical tool for quantifying skin tone.

Another limitation of our study involves potential medical conditions that could affect the accuracy of the pulse oximeters. Although our exclusion criteria accounted for many of these conditions, others, such as anemia or hemoglobinopathies, could have been present in these patients and could have affected the readings. However, several studies on patients with anemia and sickle cell anemia have concluded that, although there is some variability in the relationship between SpO2 and the true arterial saturation measurements, this was not a clinically significant difference, and that pulse oximetry should remain an important diagnostic tool in these patients [28–31].

In the future, more subjects with low blood oxygen saturation ($SpO_2 < 90\%$) should also be included in such analytical and clinical validation studies. The 90% threshold is important clinically because most people have a normal resting oxygen level over 90% and rarely go below this range unless an acute event is ongoing. For example, of patients who were infected with COVID-19 and self-monitored at home, fewer than 7% had one or more readings below 92% [32]. There were only three individuals in our study who had average actual $SpO_2$ values (as measured by the reference standard Masimo MightySat Rx pulse oximeter) lower than 90%, and therefore we were unable to evaluate whether the performance of the smartwatch $SpO_2$ measurement functionality is moderated by the actual $SpO_2$ levels. This type of study would be important to determine whether a device consistently underperforms in a critical range. For example, it is known that pulse oximeters underperform under circumstances of low perfusion [33]. This could be dangerous practically because when $SpO_2$ values are truly lower, a patient who tests positive for COVID-19 might delay seeking care as a result of overestimated measurements.

## Supporting information

**S1 Text. Indicated disclaimer and factors that are related to the $SpO_2$ detection function.**
(DOCX)

**S1 Table. Descriptive statistics of readings from clinical-grade oximeters and commercial smartwatches.**
(XLSX)

**S2 Table. Coefficients and p-value from the ordinary least square model of Fitzpatrick skin tone and metrics.**
(XLSX)

**S3 Table. Coefficients and p-value from the ordinary least square model of ITA and metrics.**
(XLSX)

**S4 Table. The number of subjects of each Fitzpatrick scale with missing ITA.**
(XLSX)

**S1 Fig. The relationship between ITA scale and Fitzpatrick Skin tone.**
(PNG)

## Acknowledgments

The content is solely the responsibility of the authors and does not necessarily represent the official views of AstraZeneca or the National Institutes of Health.

## Author Contributions

**Conceptualization:** Satasuk Joy Bhosai, Laurie Snyder, Jessilyn Dunn.

**Data curation:** Connor Spies, Justin Magin.

**Funding acquisition:** Satasuk Joy Bhosai, Laurie Snyder, Jessilyn Dunn.

**Software:** Yihang Jiang.

**Supervision:** Laurie Snyder, Jessilyn Dunn.

**Visualization:** Yihang Jiang.

**Writing – original draft:** Yihang Jiang, Connor Spies, Justin Magin, Satasuk Joy Bhosai, Laurie Snyder, Jessilyn Dunn.

**Writing – review & editing:** Yihang Jiang, Connor Spies, Justin Magin, Satasuk Joy Bhosai, Laurie Snyder, Jessilyn Dunn.

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
