## [Decision Letter · Decision Letter 0]

11 Apr 2023

PDIG-D-23-00054

Investigating the accuracy of blood oxygen saturation measurements in common consumer smartwatches

PLOS Digital Health

Dear Dr. Dunn,

Thank you for submitting your manuscript to PLOS Digital Health. After careful consideration, we feel that it has merit but does not fully meet PLOS Digital Health's publication criteria as it currently stands. Therefore, we invite you to submit a revised version of the manuscript that addresses the points raised during the review process.

Please submit your revised manuscript within 30 days May 11 2023 11:59PM. If you will need more time than this to complete your revisions, please reply to this message or contact the journal office at digitalhealth@plos.org. Please include the following items when submitting your revised manuscript:

We look forward to receiving your revised manuscript.

Kind regards,

Barret Rush, M.D.

Guest Editor

PLOS Digital Health

Journal Requirements:

2. Please send a completed 'Competing Interests' statement, including any COIs declared by your co-authors. If you have no competing interests to declare, please state "The authors have declared that no competing interests exist". Otherwise please declare all competing interests beginning with the statement "I have read the journal's policy and the authors of this manuscript have the following competing interests:"

3. Please amend your detailed Financial Disclosure statement. This is published with the article. It must therefore be completed in full sentences and contain the exact wording you wish to be published.

4. We ask that a manuscript source file is provided at Revision. Please upload your manuscript file as a .doc, .docx, .rtf or .tex.

5. Please provide separate figure files in .tif or .eps format only and remove any figures embedded in your manuscript file. Please also ensure that all files are under our size limit of 10MB.

Additional Editor Comments (if provided):

Interesting study, please see the reviwers comments below.

The study would have been much more powerful if ABG had been used as the gold standard, however with some modifications I think the paper could be published with more limitations acknoweldged.

Reviewers' comments:

Reviewer's Responses to Questions

**Comments to the Author**

1. Does this manuscript meet PLOS Digital Health’s publication criteria? Is the manuscript technically sound, and do the data support the conclusions? The manuscript must describe methodologically and ethically rigorous research with conclusions that are appropriately drawn based on the data presented.

Reviewer #1: Yes

Reviewer #2: Yes

2. Has the statistical analysis been performed appropriately and rigorously?

Reviewer #1: Yes

Reviewer #2: Yes

3. Have the authors made all data underlying the findings in their manuscript fully available (please refer to the Data Availability Statement at the start of the manuscript PDF file)?

Reviewer #1: Yes

Reviewer #2: Yes

4. Is the manuscript presented in an intelligible fashion and written in standard English?

Reviewer #1: Yes

Reviewer #2: Yes

5. Review Comments to the Author

Reviewer #1: This study compares the accuracy and reliability of SpO2 measurements from consumer smart watches to a commercially available pulse oximeter. This is an important topic as the popularity and availability of these devices is growing. I offer the following major and minor comments for consideration: 

Major comments

• A wide range of SpO2 measurements were reported (82.0-100.0%) and no explanation is offered for the range of results outside normal values. Co-morbidities such as participant history of respiratory disease should be reported, if known. Furthermore, if any of the participants were receiving oxygen therapy this should be reported. It is also unclear in the methodology if the patients were inpatients or outpatients. This would allow for interpretation of the results in the various clinical scenarios. The authors should also offer any possible explanations for the variation in oxygen saturations reported. 

• This study compares the accuracy and reliability of pulse oximetry from consumer smart watches to a commercially available pulse oximeter. Both technologies are based on the absorption of light at different wavelengths. However, it has been shows that there are racial biases in the pulse oximetry measurements, when compared to the gold standard of arterial blood gas measurements. A significant limitation of this study is that the consumer watch pulse oximeters are directly compared to another device that has known racial biases, therefore this bias is not accounted for in this study when measuring the reliability and accuracy of the smart watches. Furthermore, the accuracy of pulse oximeters is also influence by several other factors such as poor perfusion, anemia and hemoglobinopathies. These limitations should be acknowledged and discussed in the manuscript. 

• The Fitzpatrick skin phototypes is a scale that was designed to assess a person’s risk of developing sunburn, although it is often used as a proxy for race/ethnicity. The scale is disproportionally representative of White skin types. Additionally, the authors report the breakdown of included races as only Black or White race. The limitations and biases of the FST scale as well as the lack of inclusion of participants from other races should be acknowledged and discussed by the authors. 

Minor comments

• The authors should include references in the second paragraph of the introduction (lines 63-74) to support their statements about the use of pulse oximetry in COVID-19 disease and the use of home oxygen saturation monitoring. 

• Line 85-87 the authors discuss that they have recently assessed the accuracy of heart rate measurements from smartwatches. A reference should be provided for this. 

• The authors report missing data for the ITA scale for 11/49 patients. It would be helpful to report the Fitzpatrick data for the missing ITA measurements to allow readers a more comprehensive assessment of the missing data and potential biases. For example, were the missing ITA measurements predominantly one end of the spectrum of the Fitzpatrick scale?

• In figure 2, I suggest keeping the same color scheme (ie. Blue for within error range, green for underestimate) as in figure 1 for consistency between the figures.

Reviewer #2: Thank you for allowing me to review your study. This manuscript compares the accuracy and reliability of Sp02 measurements from consumer wearable smart devices to a single commercially available pulse oximeter. 

Major Comments:

Were these inpatients or outpatients or just patients recruited to a lab? It is not very clear

The authors should report co-morbid pulmonary disease if available.

The authors only use one version of a “gold” standard pulse oximeter to compare to. The “gold standard” should be an arterial blood gas. Also if you were not going to use the “Gold” standard, why just use one model of pulse oximeter, you could have used 2-3 and made an average reading. 

There are significant interference in the wavelengths by skin pigmentation in different races. A significant limitation to this study is that consumer wearables are compared to a device with known racial bias allows bias into the results which are not accounted for. If the authors had compared the wearable devices to an arterial blood gas, this bias would be taken out of the equation. This should significantly be highlighted in the limitations section. 

The Fitzpatrick skin scale is increasing being discovered to not act well in non-White skin types. The limitations of this scale for use in minority patients should be acknowledged.

6. PLOS authors have the option to publish the peer review history of their article (what does this mean?). If published, this will include your full peer review and any attached files.

**Do you want your identity to be public for this peer review?** For information about this choice, including consent withdrawal, please see our Privacy Policy.

Reviewer #1: No

Reviewer #2: No

---

## [Editor Report · Decision Letter 1]

8 Jun 2023

Investigating the accuracy of blood oxygen saturation measurements in common consumer smartwatches

PDIG-D-23-00054R1

Dear Dr Dunn,

We are pleased to inform you that your manuscript 'Investigating the accuracy of blood oxygen saturation measurements in common consumer smartwatches' has been provisionally accepted for publication in PLOS Digital Health.

Best regards,

Barret Rush, M.D.

Guest Editor

PLOS Digital Health

Thank you for answering the questions posed by the reviewers